# In Vitro and In Vivo Anticancer Activity of Basil (*Ocimum* spp.): Current Insights and Future Prospects

**DOI:** 10.3390/cancers14102375

**Published:** 2022-05-11

**Authors:** Simone Perna, Hajar Alawadhi, Antonella Riva, Pietro Allegrini, Giovanna Petrangolini, Clara Gasparri, Tariq A. Alalwan, Mariangela Rondanelli

**Affiliations:** 1Department of Biology, College of Science, University of Bahrain, Sakhir Campus, Zallaq P.O. Box 32038, Bahrain; hajaralawadhi01@gmail.com (H.A.); talalwan@uob.edu.bh (T.A.A.); 2Development Department, Indena SpA, 20139 Milan, Italy; antonella.riva@indena.com (A.R.); pietro.allegrini@indena.com (P.A.); giovanna.petrangolini@indena.com (G.P.); 3Endocrinology and Nutrition Unit, Azienda di Servizi alla Persona ‘‘Istituto Santa Margherita’’, University of Pavia, 27100 Pavia, Italy; clara.gasparri01@universitadipavia.it; 4IRCCS Mondino Foundation, 27100 Pavia, Italy; mariangela.rondanelli@unipv.it; 5Unit of Human and Clinical Nutrition, Department of Public Health, Experimental and Forensic Medicine, University of Pavia, 27100 Pavia, Italy

**Keywords:** anticancer, apoptosis, antioxidant activity, basil extract

## Abstract

**Simple Summary:**

Basil (*Ocimum basilicum*) is a medicinal herb of the family *Lamiaceae* that contains a variety of potential bioactive compounds, such as polyphenols, flavonoids, phenolics, and essential oils. *Ocimum basilicum* can boost phagocytic action of neutrophils and immunostimulant effect, antimicrobial activity due to linalool by having inhibitory action toward all tested microorganism, and additionally, rosmarinic acid shows inhibition in DNA synthesis, as well as protein synthesis when experimented on hepatoma-derived cell line (HepG2), this resulted by lower DNA fragments plus suppression on caspase-3 activation, which blocks apoptosis. The aim of this review is to spotlight and discuss the anti-cancer activity of basil (*Ocimum*) and its implications in cancer prevention and treatment. Antioxidants and other bioactive compounds in basil leaves show important potential anti-cancer activity regards to cell death and viability inhibition, cytotoxicity, inducing apoptosis, slowing down tumor growth and especially on cell cycle arrest both in vivo and in vitro.

**Abstract:**

Background: Cancer is an irregular proliferation of cells that starts with a gene mutation that alters cellular function, is triggered by several factors, and can be inherited or acquired. The aim of this review is to discuss the anticancer activity of basil and its components’ strength, focusing on its implication in cancer prevention and treatment. Methods: This systematic review involves all of the studies published from 1 January 2010 through 1 January 2022. Results: In this review, 16 research articles are included to discuss the potential anticancer ability of the extracts of various *Ocimum basilicum* varieties at various dosages, applied to different cancer cells. Of those 16 articles, 2 were in vivo studies, 13 were in vitro studies, and 1 study conducted both in vivo and in vitro experiments. Antioxidants and other bioactive compounds in basil leaves show important potential anticancer activity at dosage of 4 mg/mL as aqueous extract or essential oil up to 200 µg/mL could slow-down tumor growth and progression with regards to cell death and viability inhibition. At dosages from 50 to 500 µg/mL is effective as anti-proliferative activities. cytotoxicity, inducing apoptosis, slowing down tumor growth, and especially cell cycle arrest, both in vivo and in vitro. Human studies show effects at dosages from 1 to 2.5 mg/daily on general vital activities and on reducing cytokines activity. Conclusions: Based on 16 published studies, basil demonstrates important anticancer activities in vivo and vitro models, and it could act as a potential cancer.

## 1. Introduction

Cancer is an irregular proliferation of cells that initiates with a gene mutation that alters cellular function, disrupts cellular interactions and relations, and then leads to oncogene generation. It can be inherited or acquired [1]. Cancer cells need metabolic reprogramming to start tumor formation and progression. Cancer cells’ mechanisms for survival mostly depend on changing their own flux via metabolic pathways, because of the increased demands on bioenergetic and biosynthetic pathways, along with oxidative stress moderation [2]. This action is triggered by several factors; the International Agency for Research on Cancer (IARC) specifies external influents that can contribute to causing cancer, including alcohol, pharmaceuticals, tobacco, sunlight, etc. [3]. Detection of cancer allows a better response from the body toward any medical interventions; technical methods for the early-stage detection of cancer already exist [4].

Chemotherapy is the most frequently used therapeutic option to treat cancer. However, it also destroys many healthy cells, and is hampered by the undesirable drug resistance effect [2]. Monoclonal antibody treatment may serve as an alternative to chemotherapy due to its capacity to target specific cancer cells at the molecular level, while simultaneously promoting the induction of long-lasting antitumor immune responses [3,4]. In addition, cancer resection surgery and radiotherapy can be applied to treat cancer cells, either as a solitary approach or as a combination of both therapies [5]. Additionally, these can be used in combination with other approaches, such as immunotherapies [6,7,8,9].

In the approach to cancer therapy, one should take into consideration the use of novel agents in addition to chemotherapy, hormonal therapy, and gene-targeted therapy, together with immune-targeted approaches [10].

Generally, plants inspire scientists and specialists to develop drugs and medicines, so many are plant-derived due to their powerful chemical compounds [11]. Herbs can be used as a potential source of anticancer activity, but this process requires medical supervision, because in many cases some herbs can cause allergic reactions or gastrointestinal troubles [3]. The use of medicinal herbs for cancer treatment alongside conventional therapies has received increasing recognition due to their anticancer potential [12]. In particular, the mechanisms of herbs’ (e.g., basil) activity against cancer can include cytotoxic effect [11], cancer cell proliferation inhibition [13], efficient reduction in tumor volume, and the ability to protect DNA from threatening radiation, thus increasing survival rates [14].

Furthermore, the use of medicinal plants for cancer treatment alongside conventional therapies has received increasing recognition due to their anticancer activity [6]. The National Cancer Institute has screened about 35,000 plant samples and 114,000 extracts for potential anticancer activities [7]. The benefits of medicinal herbs in tumor treatment have been attributed to their phytochemical content [8,9,10]. Phytochemicals are compounds derived from the secondary metabolism of plants, and include flavonoids, phenolic acids, and essential oils. Such bioactive compounds have been found to be potent immunomodulators, with the potential to minimize negative impacts from cancer treatment [11].

In particular, basil (*Ocimum* spp.) is a medicinal herb of the family *Lamiaceae*, with pharmaceutical properties, and has been traditionally used to treat various diseases and conditions worldwide, including fevers, flu, cold, malaria, kidney disorders, and menstrual irregularities [12,13,14]. The species that comprise all types of basil are native to the tropical and sub-tropical regions of Asia, Africa, and Central and Southern America. Their infusions possess high antioxidant, antimicrobial, anti-inflammatory, antidiabetic, and anticancer activities [15,16,17,18]. 

Basil contains a wealth of chemical components that give it its unique taste and aroma. Essential oils in basil leaves include a wide variety of aromatic compounds such as linalool, estragole, methyl cinnamate [19], and other influential compounds, including 1,8-cineole, methyl chavicol, eugenol, bergamotene, α-cardinol, limonene, geraniol, and camphor [20]. Generally, when examining *Ocimum sanctum*, phenolics that are present in basil include rosmarinic acid, apigenin, cirsilineol, cirsimaritin, isothymusin [14], caftaric acid, chicoric acid, and caffeic acid [13], along with the flavonoids orientin and vicenin [14].

The mechanism of the aforementioned chemical compounds’ actions can be as follows: The flavonoid composition in *Ocimum basilicum* can boost the phagocytic action of neutrophils and immunostimulant effects [11]. Additionally, it is noticeable that linalool content is the highest in basil essential oils, and it has been suggested that basil’s antimicrobial activity is due to linalool exerting inhibitory action toward all tested microorganisms—particularly against Gram-positive bacteria [20,21]. Additionally, rosmarinic acid in basil (*Ocimum basilicum)* showed inhibition of DNA synthesis, as well as protein synthesis, when tested on a hepatoma-derived cell line (HepG2), resulting in lower DNA fragmentation along with suppression of caspase-3 activation, which blocks apoptosis [11].

Moreover, basil contains a wealth of chemical components that give it its unique taste and aroma. Essential oils in basil leaves include a wide variety of aromatic compounds, such as linalool, estragole, methyl cinnamate, and others [19,20].

Furthermore, the high economic value of basil oil is due to the presence of phenylpropanoids—such as eugenol, chavicol, and their derivatives—or terpenoids, such as the monoterpene alcohol linalool, methyl cinnamate, and limonene [21]. The antioxidant activity of phenolic compounds is mainly due to their redox properties, which can play an important role in absorbing and neutralizing free radicals, quenching singlet and triplet oxygen, and decomposing peroxides [22].

The amounts and activities of these essential oils vary according to genetic variability, environmental conditions, and geographical area [22,23,24]. For these reasons, the numerous *Ocimum* species’ antitumor properties are under serious investigation in order to exploit the benefits of basil extracts to develop new potential therapies. 

As an example, higher content of eugenol was found in shade-grown basil, while higher linalool content was found in basil growing without shade; this disparity was also present when the geographical growing region was different [23]. Few noteworthy reviews of basil’s anticancer activity exist; two focuses on the anticancer effectiveness of *Ocimum sanctum* (holy basil), both conducted in 2013, and another was conducted 2017, on a small scale (only four papers reviewed). For these reasons, basil’s antitumor properties are under serious investigation in order to exploit the benefits and power of basil extracts to develop new potential therapies, helped by specifying basil’s inflammation response and immunomodulation for better outcomes. The aim of this review is to spotlight and discuss the anticancer activity of basil (*Ocimum*) and its implications for cancer prevention and treatment, including mechanisms such as cell death and viability inhibition, cytotoxicity and antioxidant activity, inducing apoptosis, slowing tumor growth, and cell cycle arrest, based on the administration of different *Ocimum* varieties at specific concentrations.

## 2. Materials and Methods

### 2.1. Search Strategy

The present systematic review was conducted in accordance with the Preferred Reporting Items for Systematic Review and Meta Analyses (PRISMA) statement (registration ID: 329264). The research involved all studies published from 1 January 2010 through 1 January 2022, since—based on our research—most of the related papers and topics were published within this period. Articles published in English were identified by searching the PubMed and Scopus databases, along with manual search using Google Scholar. The search strategy was based on the following items: “basil” AND “cancer” OR “inflammation” OR “tumor” OR “proliferation”.

### 2.2. Inclusion and Exclusion Criteria

For each of the relevant abstracts, full publications were retrieved for evaluation based on criteria established a priori. English research articles published between 2010 and 2022 were included. Research articles based on in vitro and in vivo studies were included, while in-silico-based research articles were excluded. Research articles focusing on basil were included; however, those considering a mixture of plant extracts or a formulation composed of a mixture of several constituents were not considered. The relative study design and level of evidence demonstrated were as suggested by the Centre for Evidence-Based Medicine.

## 3. Results and Discussion

Basil (*Ocimum basilicum*) is a medicinal plant whose extract has proven its potential anticancer activity in many recent studies. In this review, 16 research articles were included to discuss the actual anticancer ability of the extracts of various *Ocimum* varieties. Of the 16 articles, 2 were in vivo studies, 13 were in vitro studies, and 1 study conducted both in vivo and in vitro experiments. These research articles and their main findings are summarized in Table 1, Table 2 and Table 3 and represented graphically in Figure 1. The anticancer effects were described by a number of mechanisms, such as cell death and viability inhibition, cytotoxicity and antioxidant activity, apoptosis, reduced tumor growth, and cell cycle arrest.

### 3.1. Basil and Inflammation Response

Inflammation is a natural mechanism in hosts to defend themselves, and is triggered by innate immune receptors that recognize pathogens and damaged cells [41]. Inflammation is one of the most prominent characteristic features of cancer, and plays a key role in mediating cancer initiation, cell proliferation, and cancer progression [42]. It is mainly an interaction between inflammation mediators and inflammatory cells [43]. The molecular mechanisms generally involved in suppressing inflammation include inhibiting pro-inflammatory mediators and catalyzing anti-inflammatory agents [43,44]. Several studies have discussed and demonstrated the anti-inflammatory potential of the extracts of different basil varieties [45,46]. For example, anti-inflammatory activity and anti-proliferative activity of the hydroethanolic extract of lemon basil (*O. × citriodorum*) were reported in four different human cancer cell lines (HT–144, MCF–7, NCI–H460, and SF–268) [18]. Moreover, Lantto et al. [47] reported the chemopreventive effects of water extract from basil (*O. basilicum*) on the membrane integrity, metabolic activity, and p53 protein levels of SH-SY5Y neuroblastoma cells. In addition to the decrease in metabolic activity by more than 50%, a significant decrease in the integrity of cell membranes and a significant increase in the amount of p53 in the tumor cells was observed after the exposure to 2.0 mg/mL of basil extract, as compared to the corresponding control group. Loss of p53 activity is considered to be ubiquitous to all cancers, and results in unleashed inflammatory responses due to loss of p53-mediated nuclear factor kappa B (NF-κB) suppression [48].

### 3.2. Human Studies on Immunomodulation and Potential Risks on Overdose Effects

Basil has the potential to protect cells from the different phases of cancer through its promising essential oils, including eugenol, linalool, and methyl eugenol. The application of basil extract has been shown to downregulate the expression of genes that endorse cellular propagation, migration, and angiogenesis of tumor cells, in addition to downregulating the focal adhesion kinase (FAK)-related pathway, and activating the extracellular-signal-regulated kinase (ERK)1/2 signaling pathway [49]. Moreover, the aqueous extract of basil considerably overwhelms the proliferation, motility, and invasive ability of pancreatic cancer cells. It significantly upregulates the genes that induce apoptosis and metastasis, and downregulates the genes that endorse chemo-/radiation resistance and tumor survival [50].

On the other hand, Miele et al. [51] found that the use of smaller and immature basil (*O. basilicum*) plants in the production of a typical Italian sauce called “pesto” might pose some genotoxicity concerns due to the predominant levels of the carcinogenic phenylpropanoid methyl eugenol. The authors concluded that taller basil plants with a height above 16 cm should be used for culinary purposes, as linalool and eugenol are their main components.

### 3.3. Cell Death and Viability Inhibition

Since the main result desired from any cancer treatment is to lower the viability and availability of tumor cells, many articles have highlighted the properties of the extracts of various *Ocimum* spp. varieties. In the study of Taie et al. [26], *O. basilicum* (sweet basil) leaves, their ethanolic extract, and its essential oils were applied at different concentrations to Ehrlich ascites carcinoma cells (EACC) injected into female Swiss albino mice (with body weight ranging from 22 to 25 g). This resulted in the alteration of the viability of the targeted cells compared to the controls. The authors attributed the anticancer activity to the essential oils present in the basil. The antioxidant capacity was boosted by these essential oils, and its capacity increased with increasing concentration of basil essential oils [52]. Moreover, the antioxidant capacity and activity of the basil-derived essential oils, including their enzymatic activity as free radical scavengers, may work in alignment with their anticarcinogenic actions [53]. Nonetheless, the same study is distinguished by changing the fertilization values and methods to measure their effects on the existence of the essential oils, which directly influenced the antioxidant and anticancer activities, finalized with the importance of bio-organic fertilization in growing basil plants to promote greater levels of phenolics, flavonoids, and essential oils [26]. Consequently, this resulted in better anticancer activity and a higher percentage of dead cells. 

Other articles (see Table 1 and Table 2) have also attributed the anticancer activity to the essential oil composition of basil, including the study by Gajendiran et al. [28], who used *O. basilicum* seed extract on MG63 human osteosarcoma cells to monitor the anticancer activity. They concluded that basil seeds have significant levels of bioactive components that play a major role in the cytotoxicity and deterioration of cancer cells. The minimum cell viability percentage for higher-concentrated extracts (i.e., 200 µg/mL) was observed, with cell shrinkage and membrane blebbing, indicating cell apoptosis. As such, increasing the basil seed extract concentration causes an increase in the occurrence of cell death. Furthermore, Aburjai et al. [30] concluded that the essential oils—including linalool, eugenol, and eucalyptol—in hydrodistilled *O. basilicum* (cinnamon) leaf extract have significant anticancer activity on the cancer cell lines MDA-MB-231, MCF7, and U-87 MG.

Interestingly, the study by Mahmoud [27] explored basil as a potent anticancer agent in in vitro and in vivo experiments. Specifically, marigold (*Tagetes minuta* L) and basil (*O. basilicum* L) were tested on three different types of human cancer cells, namely, promyelocytic leukemia cell lines (HL-60 and NB4), and EACC. The antioxidant activities of basil and marigold were measured by the 2,2-diphenyl picrylhydrazyl method, in light of the fact that antioxidant activities parallel anticancer activities, with both owing to the presence of essential oils. The author reported *T. minuta* as showing a higher effective concentration for 50% inhibition (EC50) value of 86.35 µg/mL, compared to 80.84 µg/mL for *O. basilicum,* which was attributed to the presence of monoterpenes in marigold. Moreover, basil was shown to contain 23 compounds, with estragole, 1,8-cineole, and linalool being the most important essential oils. In the in vitro experiment, the essential oils of marigold demonstrated a higher activity against the NB4 (81.87% dead cell percentage at 200 µg/mL) and EACC cell lines, 82.33% was the dead cell percentage for the HL-60 cell line with 200 µg/mL basil. Another method used to test the efficacy of basil and marigold as anticancer agents was by measuring the LC50 value, which suggested that the lethal concentration was 78.9 and 92.2 µg/mL of basil extract towards the HL–60 and NB4 cell lines, respectively. Although the basil extract was more powerful when compared to marigold, marigold showed better results against the NB4 and EACC cell lines. The in vivo experiment involved female Swiss albino mice, targeting EACC at three phases: pre-initiation, initiation, and post-initiation. By evaluating the percentage of dead cells, the authors demonstrated that there was a positive effect against the targeted cells, except for post-initiation for the basil group as well as the control group. Generally, the positive results of both extracts against the targeted cells were more evident in the pre-initiation groups when compared to the initiation and post-initiation group, with basil’s activity being mild in terms of toxicity. One explanation of the anticarcinogenic effects reported is the activity of the enzyme lactate dehydrogenase (LDH), with the cell membrane losing its functionality during apoptosis and/or tissue damage, thus releasing LDH from inside the cell. The increase in LDH levels indicates a decline in viability and cell numbers.

In addition to the previous study, Hanachi et al. [35] compared the anticancer effects of two herbal extracts—*O. basilicum* (sweet basil) and *Impatiens walleriana* (busy Lizzie)—on the human gastric adenocarcinoma (AGS) and human ovarian carcinoma (SKOV-3) cancer cell lines, using the MTT assay. The results of their study showed that 5 mg/mL of *O. basilicum* was effective against SKOV-3, with the half-maximal inhibitory concentration (IC_50_) of 0.91 ± 0.11 mg/mL significantly inhibiting the growth (91%), as well as increasing cell death by 63.58%. On the other hand, 5 mg/mL of *I. walleriana* promoted the growth inhibition of AGS, with the IC50 of 2.5 ± 0.21 mg/mL inhibiting cell growth by 89%, along with increasing cell death by 93.29%. Flow cytometry analysis suggested that both extracts induced apoptosis mainly due to the presence of bioactive compounds, such as caffeic acid. Caffeic acid is a potent anticancer agent known to inhibit DNA synthesis. Additionally, anthocyanins and carotenoids are important compounds present in both extracts, with *I. walleriana* having more anthocyanins. These herbal extracts provide comprehensive therapeutic benefits in medicine, with *O. basilicum* being a more common choice.

In contrast to the previous articles, Sharma et al. [36] reported no noticeable effect of *O. sanctum* (holy basil) orientin extract and its analogue on the HepG2 human liver cancer cell line. Moreover, the experimental concentration of 100 μg/mL showed no significant activity in increasing the percentage of dead cells (with only 41%) or decreasing viability with long-term exposure. Although orientin has been reported as being a potential anticancer agent due to its anti-proliferative activity [54], this review shows that orientin and its analogue fenofibryl glucuronide become non-cytotoxic against HepG2 if used as pure compounds. Furthermore, in the literature, it has been reported that orientin has a fundamental action that directs cell death, confirmed by the release of LDH [27]. In HepG2 cells, orientin also induces apoptosis and cell cycle arrest at the G_0_/G_1_ phase, through the regulation of cyclin-dependent protein kinases and their cyclin subunits [55].

Another method identified in two articles was combining basil with another medicinal plant to promote the former’s anticancer activity. The first study by Indrayudha et al. [31] found that a combination of the ethanolic extracts of *Cinnamomum burmannii* and *O. tenuiflorum* had a positive anticancer effect on T47D cancer cells. The phytochemicals in each extract exhibited a synergistic activity with cinnamon bark extract (500 µg/mL), consisting of phenols, quinones, and saponins, while the basil leaf extract (500 and 50 µg/mL) consisted of alkaloids, flavonoids, phenols, and terpenoids. The IC50 values obtained via the MTT assay were 465.21 µg/mL (*C. burmannii*) and 267.88 µg/mL (*O. tenuiflorum*)*,* with cytotoxicity augmented in the combinational extracts. In the second study, by Doguer et al. [32], different concentrations of purple basil aqueous extract in combination with sirkencubin syrup—a traditional Turkish beverage—were tested on the Caco-2 human colon carcinoma cell line. The authors found the combination to be effective as a supporting treatment to ultrasound treatment, because of the existence of aromatic bioactive compounds such as phenolic acids and flavonoids.

Another study, by Zarlaha et al. [37], used high-performance liquid chromatography (HPLC) and liquid chromatography–electrospray ionization mass spectrometry (LC/ESI-MS) to examine the effects of *O. basilicum* extract on the survival of four cancer cell lines, namely, HeLa, FemX, K562, and SKOV3. Findings revealed that the extracts exhibited high cytotoxicity against all cell lines tested, with an outstanding effect against SKOV3. They concluded that the anticancer effect of *O. basilicum* extract can be attributed to the presence of rosmarinic and caffeic acid, along with several essential oils, including eugenol, isoeugenol, and linalool. Moreover, caffeic acid exerted a powerful anti-proliferative effect by inhibiting matrix-degrading metalloproteinase-9 (MMP–9) [25]. 

### 3.4. Cytotoxicity and Antioxidant Activity

The use of AgNP nanoparticles, fabricated and synthesized with basil extract, was identified in two articles. In the first article, Abbasi et al. [38] discussed the anti-proliferative activity of sliver nanoparticles synthesized using the extract of purple basil callus (BC–AgNPs), in addition to silver nanoparticles synthesized using the anthocyanin extract of purple basil (AE–AgNPs). Generally, AgNPs have been previously reported to have the potential to cause cell death via the synthesis of reactive oxygen species (ROS) and cytotoxicity in the HepG2 cell line. Purple basil anthocyanins are common natural pigments that are among the major water-soluble flavonoids of significance. The mechanism of treating cancer was similar in each sample, with BC–AgNPs rich in secondary metabolites such as anthocyanins, caffeic acid, rosmarinic acid, and chicoric acid being the reason behind the reduction and capping of viability. However, when applied with 200 µg/mL of BC–AgNPs, the viability was only 72.49 ± 5.8%. On the other hand, AE–AgNPs in comparison with the control group represented greater cytotoxicity (75% when treated with 200 µg/mL). The authors concluded that anthocyanin-mediated nanoparticles (AE–AgNPs) are efficient in exhibiting anticancer activity because anthocyanin acts individually as a capping agent compared to BC–AgNPs. Moreover, anthocyanins have been reported to exhibit potent anticancer activity [35]. In the literature, anthocyanins have demonstrated their ability in altering some enzymatic activities that lead to cell death, such as activation of the caspase enzyme [56]. Caspases are the central components of apoptosis, and can possess tumor-suppressor functions [57]. Moreover, anthocyanins enhance chemotherapy sensitivity, consequently working as a supporter for the medical interventions in cancer therapies [56].

Another variety of basil used in the study of Złotek et al. [39] involved lettuce leaf basil (*O. basilicum*). The extract was used in the elicitation of basil with arachidonic acid (AA) in different concentrations of 0.01 mM (AA1), 1 mM (AA2), and 100 mM (AA3), to increase the extraction of phenolics such as rosmarinic, chicoric, and caftaric acids from basil extracts. Consequently, the extracted phenolics were the main source of bioactive components responsible for the anticancer activity of the basil extract. In comparison to the control group, all of the tested concentrations were shown to reduce the cellular metabolism and functionality of the human squamous carcinoma cell line SCC–15, with rosmarinic acid being the most effective. Since the anti-proliferative effect of the extract was dose-dependent, the operative doses were between 0.250 and 1.000 mg/mL. One of the mechanisms proposed to explain the apoptotic cell death and the reduction in metabolism was the suppression of lipoxygenase activity.

Alanazi [40] compared the anticancer activity of two medicinal plant species—*Ocimum* and *Achillea*—against three human cancer cell lines: a lung adenocarcinoma cancer cell line (A549), a prostate cancer cell line (PC3), and a cervical cancer cell line (HeLa). The findings of the study suggested that, at specific concentrations (41.6, 20.8, and 10.4 mg/mL), basil and yarrow extracts exhibited significant effects on HeLa cells and PC3 cells, respectively. The author attributed the cytotoxic impact of the herbal extracts to their bioactive secondary metabolite contents. Furthermore, *Achillea* extracts at lower concentrations (74.9, 37.4, 18.7, and 9.4 mg/mL) were able to lower the viability of PC3 cells, but higher concentrations (599, 299.5, and 149.8 mg/mL) were demonstrated to be effective against A549 and HeLa cells. On the other hand, *Ocimum* extract at lower concentrations (41.6, 20.8, and 10.4 mg/mL) was able to lower the viability of HeLa cells, but higher concentrations (333, 166.5, and 83.3 mg/mL) were observed to be effective against A549 and PC3 cells. Overall, the basil extract was more effective as an anticarcinogenic agent. It was suggested that antioxidants could also take on an anti-tumorigenic role, similar to the previous findings of Mahmoud [27]. 

### 3.5. Inducing Apoptosis

A recent study by Alkhateeb et al. [29] investigated the effects of *O. basilicum* (dark purple basil) blossom extract on the human breast cancer cell line MCF7. The study used low-temperature extraction (0 °C) as compared to conventional solvent extraction. The findings of their study showed that extracting at low temperatures ensured a more effective sample, with higher amounts of flavonoids and phenolic compounds in comparison to high-temperature or alcoholic extracts, thereby increasing the anticancer and antioxidant effects. The extraction temperature was identified as being one of the most significant variables influencing the extraction yield of flavonoids and other polyphenolic compounds [58]. The mechanism by which cancer cell growth was inhibited could be explained by mediated mitochondrial fission that acts to induce apoptosis in the targeted tumor cells selectively. Furthermore, the aqueous extract used showed an inhibition of 2-[N-(7-nitrobenz-2-oxa-1,3-diazol-4-yl)amino]-2-deoxy-D-glucose (2-NBDG) uptake after applying the basil blossom extract to MCF7 cells—which, in turn, affected cell growth and development negatively, with mitochondrial splitting occurring in the tumor cells and leading to irregular ATP production [29]. The authors concluded that basil blossom extract had a boosting and amplifying effect on the activation of p53, caspase-3, and DR4 mRNA activity, particularly at higher concentrations (150–250 μg/mL), inducing apoptosis and suppressing the proliferation of MCF7 cells.

In another study involving a variety of basil extracts, Elansary and Mahmoud [34] investigated different compositions with regards to their anticancer activities on several human cancer lines, so as to verify which combination had the greatest impact. All of the examined concentrations and varieties demonstrated powerful anticancer effects, with the *O. basilicum* (sweet basil) variety exhibiting significant antioxidant and anticancer action. Nonetheless, all of the basil extracts induced apoptosis and cell cycle progression, which led to anti-proliferation and cell death. It was suggested that such properties are likely to be related to the bioactive compounds, including the presence of compounds such as rosmarinic, chicoric, and caftaric acids [37,38,39].

### 3.6. Slowed Tumor Growth

One of the expected effects of *O. basilicum* extract application is the inhibition of cancer cell growth, as reported in several studies. For example, Nangia-Makker et al. [25] reported that *O. gratissimum* (holy basil) aqueous extract applied to MCF10ADCIS.COM cells injected into female nude mice resulted in the deceleration of the tumor growth. The authors suggested that the retarded tumor growth was due to suppressing MMP-2 and MMP-9 enzymatic activity, chemotaxis, and chemoinvasion. It is worth mentioning that MMP-2/-9 activity includes encouraging cell growth, cell invasion, and angiogenesis; thus, affecting this enzymatic activity this will lead to changes in cell behavior [37]. MMP-2/-9 activity is measured by knowing that basement membrane synthesis in situ is the hallmark of carcinoma, which shows a delay in formation [25]. Consequently, this indicates that the *O. gratissimum* extract can change the tumor growth via inhibition.

### 3.7. Cell Cycle Arrest

The use of nanoparticles with basil as an anticancer agent [38] was followed by the study of Manikandan et al. [33], who used AgNPs fabricated with *O. americanum* aqueous leaf extract on the A549 human lung cancer cell line to check their anticancer activity. By using fluorescent staining techniques and flow cytometry analysis, the authors concluded that the examined sample had increased cytotoxic and apoptotic activity, characterized by G_0_/G_1_ cell cycle arrest. *O. sanctum* (holy basil) has been used in traditional Indian medicine in a variety of clinical applications. 

A recent study showed that extracts of basil leaves inhibit the proliferation, migration, and invasion of cancer cells, as well as inducing apoptosis of pancreatic cancer (PC) cell lines (AsPC–1, MiaPaCa, and Capan-1) in vitro. Intraperitoneal injections of the aqueous extract of *O. sanctum* inhibited the growth of orthotopically transplanted pancreatic cancer cell lines that led to PC. This animal study showed induction of apoptosis, whereas genes that promote survival and chemo-/radiation resistance were downregulated [50].

## 4. Conclusions

In conclusion, basil extract provides a good source of bioactive compounds for cancer prevention and treatment. However, identification of the exact phytochemical compounds in different basil varieties, as well as developing their synthetic analogues, would be the desired future directions for better understating their anticancer potential and treating cancer. Moreover, human clinical trials examining the effects of these phenolic compounds in cancer prevention and treatment should be carried out in order to support the studies reviewed herein.

## Figures and Tables

**Figure 1 cancers-14-02375-f001:**
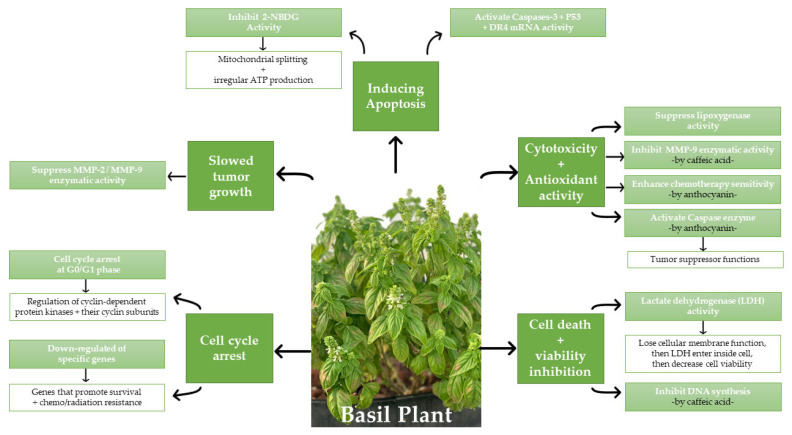
Main mechanisms and effects of *Ocimum* (basil) in immunomodulation and inflammation response.

**Table 1 cancers-14-02375-t001:** Anticarcinogenic effects of basil extracts in animal studies.

Paper	Type of Study and Level of Evidence	Compound/Extract	Sample	Posology/Treatment	Main Results
Nangia-Makker et al. [25]	Animal study (level 6)	*Ocimum gratissimum* (holy basil) aqueous extract	MCF10ADCIS.com cells injected in female nude mice	4 mg/mL lyophilized *Ocimum gratissimum*, hydrophobic or hydrophilic fractions	Slowed down MCF10ADCIS.com tumor growth and progression
Taie et al. [26]	Animal study (level 6)	*Ocimum basilicum* (sweet basil) leaves’ ethanolic extract	Ehrlich ascites carcinoma cell line injected in female Swiss albino mice 22–25 g; 8–10 weeks old	Ethanolic extracts with 1250, 1500, 1750, or 2000 ppm, with oil extracted from them (0.04, 0.06, 0.08, and 0.10 mg)	*Ocimum* ethanolic extract and oil with various concentrations changed the viability of Ehrlich ascites carcinoma cells in comparison with untreated cells
Mahmoud [27]	Animal study (level 6)	*Ocimum basilicum* (basil estragole chemotype) + fresh *Tagetes minuta* flowers (marigold) (100 g of each) were hydro-distilled	Ehrlich ascites carcinoma cell line (EACC) injected in 48 female Swiss albino mice weighting 20–25 g; 7–8 weeks old	Several volumes of marigold and basil essential oils to finalize with 25, 50, 75, 100, and 200 µg/mL concentrations	Essential oils significantly prevented tumor development (i.e., decreased total EACC number and increased the percentage of dead cells). The pre-initiation treatment was most effective compared to the initiation and post-initiation treatments, with marigold being more effective (i.e., higher percentage of dead cells)

**Table 2 cancers-14-02375-t002:** Anticarcinogenic effects of basil extracts in laboratory in vitro studies.

Paper	Type of Study and Level of Evidence	Compound/Extract	Sample	Posology/Treatment	Main Results
Gajendiran et al. [28]	Laboratory study (level 6)	*Ocimum basilicum* (basil seeds) extracted in petroleum ether and methanol separately	Human osteosarcoma cell line (MG63)	*Ocimum basilicum* seed extraction (12.5, 25, 50, 100, 200 µg/m)	Increased cell line deterioration and death with the increase in concentration of *Ocimum basilicum* seed extract.
Alkhateeb et al. [29]	Laboratory study (level 6)	*Ocimum basilicum* (dark purple blossoms of basil) aqueous extract at low temperature (0 °C)	Human MCF7 breast cancer cell line	*Ocimum basilicum* blossoms aqueous extract (0, 50, 150, and 250 μg/mL for 24 and 48 h)	Greater anticancer and antioxidant activities of *Ocimum basilicum* blossoms aqueous extract at low temperature in comparison to boiled water solvent (high temperature) and alcoholic extracts.
Aburjai et al. [30]	Laboratory study (level 6)	*Ocimum basilicum* “Cinnamon” leaves’ essential oil extracted by hydrodistillation	Three different cancer cell lines: MDA–MB–231 (triple-negative breast cancer cell line), MCF7 (breast cancer), and U–87 MG (glioblastoma)	Weight by weight (*w*/*w*%) of the dry *Ocimum basilicum* “Cinnamon” leaves was 0.50 % (*w*/*w*)	The *Ocimum basilicum* essential oils linalool, eugenol, and eucalyptol showed effective anticancer activity against several types of cancer cells.
Indrayudha and Hapsari [31]	Laboratory study (level 6)	Combination of *Cinnamomum burmannii* with *Ocimum tenuiflorum* ethanolic extract	T47D cancer cells	*Cinnamomum burmannii* ethanolic extract (500 µg/mL) + *Ocimum tenuiflorum* ethanolic extract (500 and 50 µg/mL)	Combined *Cinnamomum* and *Ocimum* ethanolic extracts produced a cooperative impact against T47D cancer cells compared to each extract alone.
Doguer et al. [32]	Laboratory study (level 6)	Purple basil (PB) dried leaves’ aqueous extract added to sirkencubin syrup (SC)	Human colon carcinoma cells (Caco-2)	Various concentrations (15–40 µL) of purple basil sirkencubin) in 100 µL of fresh medium (total of 150–400 μL/mL)	Half-maximal inhibitory concentration (IC_50_) values of SC and PBS were 288.1 and 239.8, respectively. PBS showed better results in terms of anticancer activity against (Caco-2).
Manikandan et al. [33]	Laboratory study (level 6)	*Ocimum americanum* aqueous leaf extract used to fabricate silver nanoparticles (AgNPs)	A549 lung cancer cells	Mixture of 5 g of leaf powder and 50 mL of sterile distilled water; 2 mL of the mixture treated with 100 mL of 1 mM AgNO_3_ solution	AgNPs fabricated with *Ocimum americanum* aqueous leaf extract, possessed strong cytotoxic anticancer activity against the A549 lung cancer cell line by inducing apoptosis.
Elansary & Mahmoud [34]	Laboratory study (level 6)	*Ocimum basilicum* (purple ruffle), *O. basilicum* (dark opal), *O. basilicum*. (Genovese), *O. basilicum* (anise), *O. basilicum* (bush green), and *O. basilicum* L. (OBL)	Line HeLa, MCF–7, Jurkat, HT–29, T24, MIA PaCa-2 cancer cells and one normal human cell line HEK–293	Different concentrations of the six international basil cultivars	Compounds present in *O. basilicum* species—such as rosmarinic acid, chicoric acid, and caftaric acid—varied in their anticancer activities. OBL displayed the highest antioxidant and anti-proliferative activities compared to others.
Mahmoud [27]	Laboratory study (level 6)	*Ocimum basilicum* (basil estragole chemotype) + fresh *Tagetes minuta* flowers (marigold) (100 g of each) were hydrodistilled	Human promyelocytic leukemia cell lines (HL–60 and NB4) and experimental animal model cancer cells (Ehrlich ascites carcinoma cells, EACC).	Several volumes of marigold and basil essential oils finalized to 25, 50, 75, 100, and 200 µg/mL concentrations	Dead cells increased with increasing concentrations of both estragole and marigold. Basil estragole was more effective on HL–60 than NB4 cell lines. Marigold was more effective on NB4 than HL–60 cell lines. The anticancer activity of marigold was higher than that of estragole against EACC.
Hanachi et al. [35]	Laboratory study (level 6)	*Ocimum basilicum* and *Impatiens walleriana* leaves extracts	Human gastric adenocarcinoma (AGS) and human ovarian carcinoma (SKOV–3) cancer cell lines	0.5 mg/mL to 5 mg/mL concentrations	Toxicity of *O. basilicum* on SKOV3 cell lines was higher compared to *I. walleriana*, while *I. walleriana* was more toxic towards AGS compared to *O. basilicum*. The cytotoxic effect may be attributed to the anthocyanin and flavonoid derivatives present in the extracts.
Sharma et al. [36]	Laboratory study (level 6)	Extraction of *Ocimum basilicum* L. (*O. tenuiflorum*) orientin and its analogue	Human liver cancer cell line HepG2	100 μg/mL (202.389 μM) concentration for 96 h	Cell death was only 41%, thereby indicating its ineffectiveness (in purified form) as an anticarcinogenic agent, with low cytotoxicity activity on the HepG2 liver cancer cell line.
Zarlaha et al. [37]	Laboratory study (level 6)	*Ocimum basilicum* ethanolic extract and essential oil	Human cervix adenocarcinoma HeLa cells, human melanoma FemX cells, human chronic myelogenous leukemia K562 cells, and human ovarian SKOV3 cells	Ranging from 12.5 to 200 μg/mL for 72 h (200, 100, 50, 25, and 12.5 μM)	The phytochemicals rosmarinic and caffeic acids, along with the essential oils eugenol, isoeugenol, and linalool, showed significant anticancer activity on the four cell lines—especially the SKOV3 cell line.
Abbasi et al. [38]	Laboratory study (level 6)	*Ocimum basilicum* (purple basil) callus extract used to produce silver nanoparticles (BC–AgNPs), and silver nanoparticles using anthocyanin extract derived from the same plant (AE–AgNPs)	HepG2 liver carcinoma cells	AgNO_3_ (1 mM) was added to BC (15 g callus) and AE (anthocyanin) extracts at different ratios (1:1, 1:2, 1:5, and 1:10)	AE–AgNPs showed significant anticancer activity against the HepG2 cell line (75% mortality at 200 µg/mL) compared to BC–AgNPs (approximately 27% mortality at 200 µg/mL).
Złotek et al. [39]	Laboratory study (level 6)	*Ocimum basilicum* (lettuce leaf basil) ethanolic extracts of lyophilized basil leaves	Human squamous carcinoma cell line SCC–15 (ATCC CRL1623)	Different concentrations of the extract (0.125, 0.250, 0.500, and 1.000 mg/mL)	Elicitation of basil with arachidonic acid induced overproduction of phenolic compounds that resulted in a dose-dependent decline in cell metabolism, with greater anticancer activity at higher concentrations.
Alanazi [40]	Laboratory study (level 6)	*Ocimum* spp. basil leaves and Achillea spp. extract	Human lung carcinoma cell line (A549), human prostate cancer cell line (PC3), and cervical cancer cell line (HeLa)	Basil extract concentrations (333, 166.5, and 83.3 mg/mL) for A549 and PC-3 cells, and lower concentrations (41.6, 20.8, and 10.4 mg/mL) for HeLa cells.	Basil extract worked significantly against HeLa cells, but less so against PC–3 cells. Achillea extract worked significantly against the PC–3 cell line.

**Table 3 cancers-14-02375-t003:** Immunomodulatory effects of tulsi basil (*Ocimum tenuiflorum*) extracts in the laboratory in human clinical trials.

Paper	Type of Study and Level of Evidence	Compound/Extract	Sample	Posology/Treatment	Main Results
Prasad [28]	Randomized, placebo-controlled clinical trial (level 1)	*Ocimum tenuiflorum* (tulsi) ethanolic extract (0.5%) of leaves	Healthy adults (30) aged 18–30 years	1000 mg/day for 2 weeks	Increased physical activity, while lowering increments of lactic acid. Decreased fatigue and CK levels.
Mondal et al. [38]	Randomized, double-blind, placebo-controlled crossover (level 1)	*Ocimum tenuiflorum* (tulsi) ethanolic extract of leaves	Healthy adults (22) aged 22–37 years	300 mg/day for 4 weeks	Increase in cytokine levels, especially (interferon-*ϒ* and interleukin-4).
Sharma [39]	Open clinical trial (level 1)	*Ocimum tenuiflorum* (tulsi) aqueous extract of leaves tablets	Adults with asthma (20)	500 mg × 3/day for a week	Within 3 days of relaxation, showed better vital capacity.
Rajalakshmi et al. [40]	Clinical trial (level 1)	*Ocimum tenuiflorum* (tulsi) aqueous extract of fresh leaves	Adults with viral hepatitis (20) aged 10–60 years	10 g/day, for 2–3 weeks, depending on the severity of the case	Enhancement of all symptoms within 2 weeks.
Das et al. [41]	Randomized, parallel-controlled clinical trial (level 1)	*Ocimum tenuiflorum (tulsi)* aqueous extract of fresh leaves	Adults with viral encephalitis (14)	2.5 g × 4/day for 4 weeks	In comparison with steroids, the survival rate was boosted when the examined extract was applied.

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
