# Peer review of "In Vitro and In Vivo Anticancer Activity of Basil (Ocimum spp.): Current Insights and Future Prospects"

_cancers, 2022, doi:10.3390/cancers14102375_

Round 1
Reviewer 1 Report
In this manuscript, the authors summarized an overview on the anti-cancer activity of Basil. However, there are deficiencies in this manuscript. Therefore, this manuscript is not recommended for publication in its present form, but may be reconsidered as a new paper after throughout revisions.
- The relevance of the topics covered in this review to the work of the authors to date should be mentioned in the introduction section.
- The authors need to mention in more detail whether a review article similar to the content of this manuscript was previously published. That will be useful information for the readers of this journal.
- This manuscript covers articles published between January 1, 2010 and January 1, 2022. The authors should mention why it is limited to this period. Hasn't a paper on this topic been published before that? The authors should search the literature extensively on this topic.
- The authors should also summarize in detail previous reports on the chemical constituents contained in Basil (Ocimum basilicum). Then, anti-cancer activity of Basil should be discussed. Also, the structural formula of the compound should be added.
- The authors summarize the biological activity of Basil (Ocimum basilicum) in Figure 1. However, this is too simple. It should be revised to a more detailed and easy-to-understand diagram.
Author Response
In this manuscript, the authors summarized an overview on the anti-cancer activity of Basil. However, there are deficiencies in this manuscript. Therefore, this manuscript is not recommended for publication in its present form, but may be reconsidered as a new paper after throughout revisions.
- Thanks a lot for your suggestions and valid comments and appreciation.
(1) The relevance of the topics covered in this review to the work of the authors to date should be mentioned in the introduction section.
- We included an ad hoc paragraph discussing the relevance of the topic
(2) The authors need to mention in more detail whether a review article similar to the content of this manuscript was previously published. That will be useful information for the readers of this journal.
Done, included
(3) This manuscript covers articles published between January 1, 2010 and January 1, 2022. The authors should mention why it is limited to this period. Hasn't a paper on this topic been published before that? The authors should search the literature extensively on this topic.
- All related papers are published on this period, we mentioned into the method section.
(4) The authors should also summarize in detail previous reports on the chemical constituents contained in Basil (Ocimum basilicum). Then, anti-cancer activity of Basil should be discussed. Also, the structural formula of the compound should be added.
- The chemical constituents contained in Basil (Ocimum basilicum) and anti-cancer activity of Basil has been discussed.
(5) The authors summarize the biological activity of Basil (Ocimum basilicum) in Figure 1. However, this is too simple. It should be revised to a more detailed and easy-to-understand diagram.
- Done.
Reviewer 2 Report
The abstract should better represent the review and aim of the manuscript.
The introduction is too poor. Authors should improve this part by adding more information on the state of the art in the use of medicinal herbs against cancer (technical and regulatory information). I think it may be very important to add value to this manuscript.
Table 1 can be improved by adding the main bioactive compounds in the considered extracts involved in the anti-cancer activities for each cited publication.
Conclusions should be improved in order to summarize the main issues of the review avoiding redundant information.
Author Response
(1) The abstract should better represent the review and aim of the manuscript.
- 1)A: We made additional editing to the abstract to better represent the whole manuscript
(2) The introduction is too poor. Authors should improve this part by adding more information on the state of the art in the use of medicinal herbs against cancer (technical and regulatory information). I think it may be very important to add value to this manuscript.
- 2) A: the introduction has been improved generally, mentioning chemical component and its specific effect.
(3) Table 1 can be improved by adding the main bioactive compounds in the considered extracts involved in the anti-cancer activities for each cited publication.
- 3) A: We reported into the table 1 all information reported into the main papers. Missing info are related to the missing into into the paper.
(4) Conclusions should be improved in order to summarize the main issues of the review avoiding redundant information.
- 4) A: we improved the conclusion in according to the
Reviewer 3 Report
In this manuscript the authors conduct a literature search on the anticancer activity of basil (Ocimum basilicum), focusing specifically on 16 articles with in vivo and in vitro studies detailing the effect of basil on different types of cancer. In reading the manuscript I am presented with the following comments.
Introduction: line 31-62: this part of the introduction should be revised. I suggest rewriting this part with more focus on the information to be provided. Also, in general, the number of bibliographic references consulted should be increased.
As this is an article focused on the anticancer effect of Ocimum basilicum, the different sections should be related to the benefits for cancer treatment. For example Basil and the inflammation response is an appropriate section if you want to address the therapeutic effects of this plant but do not relate to its usefulness in cancer.
Human studies on immunomodulation: this section also does not focus on cancer. It discusses various studies and immunomodulation against various infections.
Line 252-278: I do not understand why these studies are in section 3.2 and not in section 3.1.
Although the title indicates that the article evaluates the activity of Ocimum basilicum, other Ocimun varieties are mentioned, so I suggest modifying the title.
The conclusions section should be improved.
Author Response
in this manuscript the authors conduct a literature search on the anticancer activity of basil (Ocimum basilicum), focusing specifically on 16 articles with in vivo and in vitro studies detailing the effect of basil on different types of cancer. In reading the manuscript I am presented with the following comments.
Introduction: line 31-62: this part of the introduction should be revised. I suggest rewriting this part with more focus on the information to be provided. Also, in general, the number of bibliographic references consulted should be increased.
- The section has been revised and additional citations were included as suggested by the reviewer.
As this is an article focused on the anticancer effect of Ocimum basilicum, the different sections should be related to the benefits for cancer treatment. For example Basil and the inflammation response is an appropriate section if you want to address the therapeutic effects of this plant but do not relate to its usefulness in cancer.
- The section has been revised to highlight the fact that inflammation is one of the hallmarks of cancer and plays a key role in mediating cancer initiation and progression.
Human studies on immunomodulation: this section also does not focus on cancer. It discusses various studies and immunomodulation against various infections.
- The section has been revised to focus on cancer as suggested by the reviewer.
Line 252-278: I do not understand why these studies are in section 3.2 and not in section 3.1.
- We agree with the reviewer in that the first two paragraphs belong to the section related to “Cell death and viability inhibition”.
Although the title indicates that the article evaluates the activity of Ocimum basilicum, other Ocimun varieties are mentioned, so I suggest modifying the title.
- The title has been modified to better reflect as suggested by the reviewer.
The conclusions section should be improved.
- The conclusions section was modified as suggested.
Round 2
Reviewer 1 Report
This revised manuscript has been modified according to the reviewer’s comments. It is acceptable for publication.
Reviewer 3 Report
After reviewing the corrections incorporated by the authors and viewing the authors' responses, I believe that the manuscript has improved in organization and comprehensibility.